# Parental Rejection and Adolescents’ Learning Ability: A Multiple Mediating Effects of Values and Self-Esteem

**DOI:** 10.3390/bs13020143

**Published:** 2023-02-08

**Authors:** Lili Lan, Xiaofeng Wang

**Affiliations:** 1School of Marxism, Fudan University, Shanghai 200433, China; 2School of Business Administration, Shanghai Lixin University of Accounting and Finance, Shanghai 201620, China; 3Counseling and Psychological Services Center, Shanghai University of Political Science and Law, Shanghai 201701, China

**Keywords:** Chinese adolescents, parental rejection, fashion values, self-esteem, learning ability

## Abstract

Today’s society has been paying increasing attention to the important impact of family parenting practices on the development of adolescents. Adolescents with poor parenting practices may have poor academic performance in school, have low self-evaluation, and are more likely to be captured by video games and short videos. The present research explored the mediating role of fashion values and self-esteem in the relationship between a negative parenting style and adolescent learning ability. We aimed to deepen our understanding of the relationship between family parenting and adolescent value identity, as well as between individual self-esteem and school adaptation. We based our research on a total of 997 students in Shanghai from grades 6, 8 and 10. Furthermore, we made use of parenting behaviour, Chinese adolescent values, and of the revised Chinese version of the class drama questionnaires and of the Children’s self-awareness scale. The chain mediation model was used to analyse the mediating effect of fashion values and self-esteem on parental rejection and peer evaluation learning ability. The results showed that fashion values played a partial mediating role between parental rejection and adolescent learning ability, and that parental rejection positively predicted fashion values, and fashion values negatively predicted learning ability. Self-esteem played a partial mediating role between parental rejection and adolescent learning ability, while parental rejection negatively predicted self-esteem and self-esteem positively predicted learning ability. Fashion values and self-esteem played a chain mediating role between parental rejection and adolescent learning ability, as parental rejection negatively predicted fashion values, fashion values positively predicted self-esteem, and self-esteem positively predicted learning ability. In conclusion, a negative parenting style influenced the development of adolescent value recognition and self-esteem, and affected the development of adolescent learning ability. That said, we should encourage families to adopt a positive parenting style and adolescent quality education to positively impact adolescent development.

## 1. Introduction

Over the past four decades, China has become a highly competitive society, as a result of its political and economic reform. Therefore, comprehensive school skills, such as an initiative-taking approach and self-confidence, have become requirements for adaptability and success; this is recognised by parents, educators, and other social factors [1,2]. The Ministry of Education of China has modified its educational policies to follow a more comprehensive development in academic, and various other extracurricular, activities [3]. The latest national education policy, “Opinions on Further Reducing the Burden of Homework and Off-Campus Training for Students in Compulsory Education” (released in July 2021 referred to as “Double Reduction”), proposed to effectively reduce the excessive burden of homework and off-campus training for students, and promoted students’ overall development and healthy growth [4]. For that reason, families, educators, and communities are placing emphasis on the academic performance of adolescents, but also on the formation and development of their learning abilities as these can better reflect their motivation, learning habits and learning potential in the future. As adolescents grow in a media technology-infused environment, learning resources and learning channels become more available to them; thus, paying attention to and investigating the key factors that affect the development of adolescent learning ability is important.

The various parenting styles reflect the perceptions, attitudes, emotions, and behaviours of the parent in the process of raising a child [5,6]. Since family is an immediate microsystem that impacts adolescent development, it is the parents who most closely interact with adolescents. Parenting styles also have a significant influence on adolescent development in terms of learning engagement, academic behaviours, and academic attitudes [7], and poor parenting styles can negatively predict adolescent learning behaviour and academic performance [8]. However, as the involved mechanisms still need to be explored further, the present study focused on the mediating role of values and self-esteem in the relationship between poor parenting styles and adolescent academic ability.

Adolescence is a critical period for the formation and development of individual values and self-esteem [6,9,10]. Individual values and self-esteem are strongly associated with parenting styles and adolescent academic abilities [1,11,12]. It has also been observed that previous studies have not placed much emphasis on the ways that poor parenting styles can impact the values of adolescents, and thus their learning ability. Nowadays, adolescents tend to relate to the values of self-orientation, comfort, and entertainment more than the previous generations did [13]. The prompt development of the social economy and mobile network technology after China’s reform and expansion has incentivised adolescents to more keenly buy fashion products; this is because accessing information about fashion trends is much easier compared to the previous years. In this social environment, adolescents further relate to fashion, which, nowadays, has resulted in an impact upon the psychosocial adaptation and school performance of adolescents to a great extent [11]. A close relationship to fashion values brings adolescents closer to the latest clothing and electronic products. It also encourages peer interaction, as well as their ability to interact socially. In this way, it also has a positive effect on their self-esteem and promotes their development [14]. On the contrary, the more adolescents value fashion, the more they will pay attention to the comparison of wealth, appearance, and social status. This comparison would result in anxiety, depression, and other negative feelings, which can hinder healthy development [15].

The present study has been based upon previous studies that discussed how poor parenting styles have a negative impact on adolescent learning ability by reducing their self-esteem. In addition, it will explore the role of fashion values in the negative impact of poor parenting styles on adolescent learning ability. By exploring the role of fashion values in this context, we will obtain a better understanding of the relationship between family parenting and adolescent value identity, individual self-esteem, and school adjustment. What is more, the findings will provide useful insights into the family education and educational values of adolescents.

### 1.1. The Mediating Effect of Fashion Value

Values are abstract life goals, reflecting what is most important in people’s lives. Values can guide behaviours, and form a decisive evaluation of people, events, and the self [16,17]. Values are important factors that influence adolescent learning ability [1], while intrinsic and self-transcendent values (i.e., universalism and benevolence) can enhance academic motivation and improve learning strategies [18]. In contrast, extrinsic values, such as materialism, reduce intrinsic motivation and can lead to poor academic performance [19]. Amongst these values, fashion values also have a negative impact on adolescent academic performance, as they negatively predict academic performance and positively predict learning problems [11].

Over the past 40 years, during China’s reform and expansion, the values of adolescents have changed significantly [13,20]. In addition to the traditional values of, for example, family affection, adolescents also relate to modern values, such as fame and fashion values [11]. For example, Chinese adolescents believed that acquiring and owning the most popular clothing and electronics is essential, and that satisfaction and happiness in life derive from these items and from their attention to fashion trends [21]. What has led adolescents to relate to fashion values that much? Previous studies have shown that families with weak emotional ties, such as cases of parental divorce, or interparental hostility or conflict, can negatively predict adolescent social competence [22]. The excessive attention of a parent to material wealth will also motivate children to emphasise their self-value and sense of what is significant in life through the ownership of material wealth and the consumption of material products [19,23]. In summary, adolescents who do not receive suitable care and support from their families tend to seek temporary pleasure, self-esteem, friendship and security in fashion information, clothing, electronics, and popular subcultures. Furthermore, research showed that individuals who centred on acquiring material possessions had lower levels of academic engagement [15], and longitudinally had worse exam performance [24].

Therefore, we assumed that this results in adolescents identifying more with fashion values, which affected their learning ability in a negative way.

### 1.2. The Mediating Effect of Self-Esteem

Self-esteem refers to an individual’s emotional experience and evaluation of self-worth formed in the process of socialisation [6]. Adolescence is an important stage in the development of individual self-esteem. In addition, the sense of self-esteem, as an important psychological resource and competence, is closely related to adolescent psychological health and social adjustment [25]. Some studies have shown that individuals with a high self-esteem have a better academic performance and that their academic competitiveness tends to be stronger compared to others with a lower sense of self-esteem [26]. Caring parents can also promote the development of self-esteem in adolescents [27], while parental conflict is associated with the self-esteem of adolescents in a negative way [28]. Thus, we assumed that poor parenting can impact adolescent self-esteem in a negative way while the sense of high self-esteem can have a positive impact on their learning ability.

### 1.3. Chain Mediating Effects of Fashion Values and Self-Esteem

Studies have shown that individuals tend to enhance their self-esteem through the pursuit of material wealth and their relation to fashion [29]. With the development of online and mobile technologies, mainstream media, and websites, subcultural communities are no longer just tools for information exchange, but have evolved into important cognitive environments for the growth of adolescents [30]. Adolescents follow other peers, as well as gain their attention through WeChat friend circles and short videos. Based on that observation, we assumed that there was a positive outcome from the relation of adolescents to fashion values, being used as a way of improving their self-esteem, i.e., the more they relate to fashion values, the higher their self-esteem was.

### 1.4. The Present Study

Our study was a cross-sectional data study with poor parenting practices as the independent variable, peer-rated academic performance as the dependent variable, and fashion value and self-esteem as mediating variables. This study will establish multiple mediating models that derive from fashion values and self-esteem. The purpose of this study was to deepen the understanding of the influence of adolescents’ value identity and self-esteem upon family parenting and thus on learning ability. The study had the following three hypotheses: (1) fashion values play a mediating role in relation to poor parenting practices and learning ability; poor parenting practices positively predict fashion values, and fashion values negatively predict learning ability; (2) self-esteem plays a mediating role in relation to poor parenting practices and learning ability; poor parenting practices negatively predict self-esteem, and self-esteem positively predicts learning ability; and (3) fashion values and self-esteem play a chain mediating effect in relation to poor parenting practices and learning ability; poor parenting practices positively predict fashion values, fashion values positively predict self-esteem, and self-esteem positively predicts learning ability.

## 2. Participants and Methods

### 2.1. Participants and Procedure

The participants of the study were 6th, 8th, and 10th grade students from four junior high schools and three high schools in Shanghai. Participants were selected using convenience sampling, with participants coming from several middle and high schools with which the researcher collaborated; these schools were not exceptional. The students in the schools represented the general situation of the junior and senior high school students in Shanghai. In these schools, the researcher randomly selected one-third of the students in the class at different grades as participants. A total of 997 valid questionnaires were collected, including 425 students (221 boys) in grade 6th (average age = 12.93), 269 students (158 boys) in grade 8th (average age = 14.72), and 298 students (126 boys) in grade 10th (average age = 16.85). The questionnaires were distributed in the classroom and filled out by the students, and their questionnaires were collected on the spot by the researcher after they were completed. The research content and process were approved by the school leaders and the head teachers, and the students collected informed consent about the research content and the process. Shanghai is one of the regions with better economic development in China, with nearly 25 million permanent residents, half of whom are migrants. These migrants come from all provinces of China, as well as other countries. Therefore, when sampling in Shanghai, the sample data can reflect the situation of adolescents in Shanghai and other cities in China to some extent. In addition, the age span (11–17 years old) of the participants selected for the sample can also represent the age stage of adolescents to a great extent.

### 2.2. Measures

#### 2.2.1. Parental Rejection

In the present study, we used the parenting behaviour questionnaire as described by Chen [31]. The questionnaire consisted of twenty items, including four components: father/mother acceptance and father/mother rejection. The father/mother rejection component (i.e., *“My father often reprimands and criticises me”*) was used in this study, and scores were kept on a 5-point scale; with the average component scores that were calculated, higher scores indicate an increase in parental rejection behaviours. As the correlation values between the paternal and maternal rejection were high (*r* = 0.52 *p* < 0.001), we averaged the scores of the paternal rejection and maternal rejection components as one component; this was parental rejection. The internal consistency coefficient (Cronbach’s α) of parental rejection was 0.85.

#### 2.2.2. Fashion Values

In addition, we made use of the self-reported questionnaire on Chinese adolescent values. This consists of eight components: social equality, collective responsibility, rule-abiding, family wellbeing, friendship, self-improvement, fashion, and personal happiness. The total items counted were forty-six [21]. In this study, we used the fashion values component, which consists of six items (i.e., *“He/she believes that young people’s lives should be in line with the trends of social fashion”*) and used a 5-point scale to calculate the average score of the components. The higher the score, the more the individual relates to this value. The Cronbach’s α for fashion values in this study was 0.89.

#### 2.2.3. Self-Esteem

To measure the self-esteem levels of each participant, we used the Self-Perception Profile for Children [32]. The scale consists of thirty-six items, scored on a 5-point scale, and contains 6 components, including overall self-esteem and academic self-esteem, etc. In this study, the component of overall self-esteem was chosen, with six items (i.e., *“I am confident in myself”*), and higher average scores indicate higher levels of self-esteem. The Cronbach’s α for the overall self-esteem component was 0.81.

#### 2.2.4. Learning Ability

What is more, we measured adolescent learning ability using a Chinese version of the Class Drama Scale [33]. Students were asked to respond on a 5-point scale, ranging from 1 (not at all true) to 5 (always true), to thirty-five items of behaviour descriptions (i.e., *“he/she is a good leader”)*; the participants were required to nominate three classmates who best met the behavioural description. That is, learning ability in this study included three items: reading ability (i.e., *“he/she has a great reading ability”*), mathematical ability (i.e., *“he/she has a great mathematical ability”*) and academic achievement (i.e., *“he/she always knows the correct answer when the teacher asks questions”*). The score of each student’s item was calculated and standardised in the class. As the correlation coefficients of the reading ability, mathematical ability and academic achievement were very high, the results showed that reading–mathematics = 0.45 (*p* < 0.001), reading–achievement = 0.70 (*p* < 0.001) and mathematics–achievement = 0.74 (*p* < 0.001). Thus, we took the average of the three scores as the learning ability. The Cronbach’s α coefficient of learning ability was 0.92.

### 2.3. Data Processing

The study used SPSS 21.0 software for data processing. We first performed the descriptive statistical analysis, followed by the chain-mediated model analysis using Process 3.0 model 6. The Harman one-way method was used to test the common method bias, and we also performed an unrotated principal component factor analysis on all the items of the variables. A total of thirteen factors with a characteristic root greater than one were extracted, and the variance explained by the first factor was 14.06%, which was much lower than 40%. Thus, it can be concluded that there was no serious common method bias problem in this study.

## 3. Results

### 3.1. Descriptive Statistics

Table 1 indicated the averages and standard deviations of the variables, as well as the correlation coefficients of the variables. The results showed that parental rejection was significantly positively correlated with fashion values, and negatively correlated with the self-esteem and the learning ability of the participants. The fashion values were significantly positively correlated with the self-esteem of the participants, and significantly negatively correlated with their learning ability. In addition, self-esteem was significantly positively correlated with their learning ability.

### 3.2. The Relationship between Parental Rejection and Learning Ability: Multiple Mediatingi Model

All the variables were standardised, except for gender and grade. Parental rejection was used as an independent variable, whereas the learning ability variable was seen as a dependent one. Fashion values and self-esteem were mediating variables. The multiple hierarchical regression analysis was conducted according to Model 6 in Process program 3.0, allowing an integration test on the chain mediation model.

As shown in Table 2, under the control of grade and gender, the coefficient of the path of parental rejection to fashion values was significant (β = 0.29, *p* < 0.01), and the path coefficient for parental rejection to self-esteem was significant (β = −0.24, *p* < 0.01). The path coefficient for fashion values to self-esteem was significant (β = 0.17, *p* < 0.01), and the path coefficient from fashion values to learning ability was also significant (β = −0.29, *p* < 0.01). The coefficient of the path from self-esteem to learning ability was significant (β = 0.44, *p* < 0.01). After adding fashion values and self-esteem to the test, the effect of parental rejection on learning ability was no longer significant. The model plot was shown in Figure 1.

Furthermore, we used the Bootstrap method to calculate 95% confidence intervals for each of the 5000 repeated samples, and the results can be seen in Table 3. The results showed that none of the confidence intervals corresponding to the paths tested contained a 0 value. The direct effect of parental rejection on the adolescent learning ability was −0.08, and the total indirect effect (i.e., the sum of the three mediated path effect values was −0.17). As the amount of direct effect was negative in the chain mediation model, the absolute value of the ratio of indirect effect to direct effect |ab/c′| was reported in this case. The total mediating effect and the |ab/c′| for the three mediating paths were 1.88, 0.89, 1.22, and 0.22, respectively.

## 4. Discussion

This study examined the effect of parental rejection on adolescent learning ability and its underlying mechanisms. The results showed that parental rejection was significantly and negatively related to adolescent learning ability. In addition, when the two mediating variables of fashion values and of self-esteem were included in the test, the direct effect of parental rejection on learning ability became insignificant, but the adolescent learning ability was impacted even further through the separate mediation of fashion values and self-esteem, as well as the chain mediation of fashion values and self-esteem.

This study found that fashion values partially mediated the relationship between parental rejection and adolescent learning ability, as was found by testing hypothesis 1. The study revealed that parental rejection, by further encouraging the close relation of adolescents to fashion values, may subsequently hinder the development of their learning ability. A reason for this would be that adolescents who grow up in rejection-oriented families are not able to satisfy their basic psychological needs (i.e., autonomy, relational needs, and competence needs). This is due to chronic parental negative response, estrangement, and neglect. According to the self-determination theory (SDT), children and adolescents whose basic psychological needs are not met will pursue alternative methods of satisfaction from external sources [34,35]. This is because they can at least partially deal with negative emotions related to self-insecurity [36]. In the present study, adolescents who experienced parental rejection looked for satisfaction in external substances (i.e., trendy clothing and electronic products). Therefore, internalising fashion values and excessively following a fashion trend can be distracting for adolescents. This can have an impact on their learning abilities by negatively affecting the development and improvement of their learning abilities.

This study also found that a high sense of self-esteem can partially mediate the relationship between parental rejection and adolescent learning ability, as is seen by validating hypothesis 2. The sociometer theory of self-esteem suggests that self-esteem is an important indicator of an individual’s interpersonal relationships. When an individual feels accepted and appreciated by others, the individual’s self-esteem level rises. However, when an individual feels rejected by others, this leads to a decrease in their self-esteem levels [37]. Therefore, if adolescents often feel rejected when interacting with their parents, this will increase their sense of negative self-evaluation and subjective feelings, and decrease their self-esteem levels. In addition, when self-esteem, an important psychological resource, is at a low level, individuals will adopt more negative behaviours. For example, when encountering difficulties and problems in the learning process, individuals with low self-esteem are more inclined to believe that they are not capable of dealing with the problem; thus, they more often use emotional coping styles, such as avoidance and self-blame [38]. As a result, the development of their learning abilities is hindered.

The findings of this research also revealed that fashion values and self-esteem mediate between parental rejection and adolescent learning ability. The study showed that materialism is not always harmful when material wealth is used as a tool to enhance self-esteem by satisfying basic needs [39]. The above study speculated that fashion values negatively and positively affected individual psychosocial adjustment. Popular electronics and trendy clothes are important objects that help adolescents to maintain a positive perception of themselves. Adolescents’ identification with and pursuit of these trends may be seen as a substitute for the absence of basic psychological needs and the motivations that are triggered by parental rejection. Thus, they can temporarily satisfy their self-esteem and their sense of efficacy. The results of this study presented both the negative and positive effects of fashion value on adolescent development. This also presented the face of the materialistic values theory debate, i.e., whether identification with materialism can be seen as a form of self-escape or as self-identification [40]. Our results suggested that an identification with fashion values can be seen as both a form of self-escape (escaping from the home and school environment) and of self-identification (identification with one’s autonomy, belonging, and uniqueness).

In summary, this study more comprehensively presented the negative impacts of poor parenting practices on the development of adolescents. These impacts included not only the adaptation of adolescents in school (the reduced learning ability of peer evaluation), but also low self-evaluation and the desire of adolescents for fashionable clothing, electronic products and an attractive appearance over other important life goals (such as self-improvement and collective responsibility). This proves that the family is an important micro-environment that not only contributes directly to the development of the adolescent, but also serves as an individual–school and individual–society connection. Through this linkage, the family exerts a more complex and comprehensive influence on the development of adolescents.

## 5. Limitations and Implications

Some limitations of this study should be noted. First, the current sample was recruited from only in Shanghai, thus the generalization of these findings in China may be limited. For example, in areas where filial piety is more accepted, children may be more accepting of their parents’ criticism. Second, given the nature of the cross-sectional data study, the causal role of these factors could not be inferred, and the findings should be interpreted with caution.

Future studies will consider using longitudinal studies to examine the long-term effects of parental rejection on several adolescent developmental variables in the current study, in addition to increasing the diversity of participants. In addition, intervention studies will be designed, for example, to educate and support parents by conducting parenting workshops or parenting groups to improve parenting practices and reduce their neglect and criticism of their children. Then, whether their children’s developmental indicators have improved will be analysed.

Despite these limitations, the results of this study provided some insights into family education and educational values for adolescents. According to these, parents should be able to create a positive family atmosphere for their children and adopt positive parenting styles, such as encouragement and support for their children in their education. This study also found that fashion values can promote self-esteem and have an indirect positive effect on learning ability, but the excessive pursuit of fashion can still hinder the development of adolescent learning ability. Therefore, parents and educators should pay attention to guiding and intervening in adolescent values. Adolescents have a stronger sense of autonomy and can make their own choices outside of home and school. However, parents and educators should still help them understand the short-term benefits and long-term damages of agreeing with the fashion values. We should also help them realise that when they are frustrated at home and school, they can also gain other support and improve their abilities to solve problems by identifying with other values, such as collective responsibility and self-improvement. Schools can offer courses and lectures on these values to enhance adolescents’ awareness of them. This will help adolescents to analyse the values behind their behaviours. Some values (such as materialism and fashion) may be violating their accepted life values. Knowledge about the theory of these values enables adolescents to think more deeply and comprehensively about the human value system and gradually build their own value system.

## Figures and Tables

**Figure 1 behavsci-13-00143-f001:**
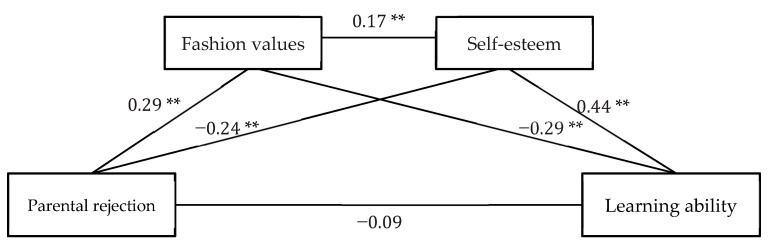
Multiple mediating model of parental rejection and learning ability. Notes ** *p* < 0.01.

**Table 1 behavsci-13-00143-t001:** Means, standard deviations and correlation between the variables.

Variables	1	2	3	4	5	6
1 Gender	-					
2 Grade	0.08 *	-				
3 Parental rejection	−0.09 **	0.02	-			
4 Fashion value	0.01	−0.03	0.25 **	-		
5 Self-esteem	−0.13 **	−0.15 **	−0.16 **	0.12 **	-	
6 Learning ability	−0.01	0	−0.09 **	−0.10 **	0.16 **	-
M	-	-	2.28	2.61	3.55	0
SD	-	-	.88	1.10	0.79	2.56

Notes * *p* < 0.05, ** *p* < 0.01.

**Table 2 behavsci-13-00143-t002:** Multiple mediating model between parental rejection and learning ability (*N* = 997).

Variables	Fashion Values			Self-Esteem			Learning Ability		
	β	SE	t	β	SE	t	β	SE	t
Constant	−0.57	0.19	−3.00 **	1.53	0.19	8.11 **	−0.08	0.51	−0.16
Gender	0.06	0.06	0.97	−0.29	0.06	−4.70 **	0.05	0.16	0.33
Grade	−0.02	0.02	−1.16	−0.08	0.02	−4.59 **	0.03	0.05	0.68
Parental rejection	0.29	0.04	8.08 **	−0.24	0.04	−6.72 **	−0.09	1.00	−0.89
Fashion values				0.17	0.03	5.36 **	−0.29	0.08	−3.47 **
Self-esteem							0.44	0.08	5.30 **
*R* ^2^	0.06			0.09			0.04		
*F*	22.14 **			23.52 **			8.56 **		

Notes ** *p* < 0.01.

**Table 3 behavsci-13-00143-t003:** Direct and mediated effects between parental rejection to learning ability.

Effect	Effect Value (SE)	|Effect Value/Direct Effect|	95% Confidence Interval
Direct effect	−0.09 (0.10)		−0.27—0.10
Path1	−0.08 (0.02)	0.89	−0.14—−0.04
Path2	−0.11 (0.03)	1.22	−0.17—−0.06
Path3	0.02 (0.01)	0.22	0.01—0.04
Total mediated effect	−0.17 (0.04)	1.88	−0.25—−0.10

Note: path1: Parental rejection—fashion values—learning ability; path2: Parental rejection—self-esteem—learning ability; path3: Parental rejection—fashion values—self-esteem—learning ability.

## Data Availability

The datasets generated for this study are available on request to the corresponding author.

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
