# Peer review of "Parental Rejection and Adolescents’ Learning Ability: A Multiple Mediating Effects of Values and Self-Esteem"

_behavsci, 2023, doi:10.3390/bs13020143_

Round 1

Reviewer 1 Report

A good article about a topical problem not only for China, but for all those interested in improving adolescents’ learning process in high school and university, especially in an age more and more dominated by aggressive technology and by laxer parental interaction.

In my opinion the abstract promises more than the article actually delivers in terms of the Chinese situation. The authors state that they “made use of parenting behaviour, Chinese adolescent values, and the revised Chinese version of the class drama questionnaires and of the children's self-awareness scale.” But the article does not show concrete examples of the above, except for general issues as for example on p. 3, par. 2 or  on p.7 under the heading “Discussion” the authors give examples of a universal adolescent behavior and quote a study relating to the USA (Deci & Ryan, 2010) as relevant and transferable to the Chinese situation as well.

The article has a quantitative study which is well presented and explained, but not entirely and explicitly related to the conclusions. The study ends with what might be called “recommendations”: “parents and educators should pay attention to identifying, guiding, and intervening in adolescent values. Schools can offer lectures and courses on the topic of educational values to enhance adolescents' awareness of values.”

I suggest a more evident transfer from the study to the conclusions/recommendations.

Also, the discussion about values starts from a relatively old article of Schwartz & Bilsky, 1987, indeed a classic, but who analyzed mature individuals and tested their theory on people from Israel and Germany. There are, however, numerous articles published in 2022 who discuss adolescent values as well as the impact of technology and fashion on their learning capabilities. My suggestion is a more updated reference list.

Some minor language editing is necessary as in “The surveys were shared and filled-in in the classrooms and the questionnaire was collected on the spot.” (p. 4)

Reviewer 2 Report

Thank you for giving me the opportunity to act as a reviewer for this manuscript.

The aim of the study is to deepen our understanding of the relationship between family parenting and adolescent value identity, as well as between individual self-esteem and school adaptation.

The manuscript is written in a simple and clear way, both in its theoretical and methodological basis. I found that the topic of the manuscript is interesting and relevant, and I think it is a timely and novelty research. However, after reading the manuscript, I ask the authors to consider the following suggestions:

  • It should be advisable to include more recent references. At least these ideas should be supported by more up-to-date references:

    • Introduction, first paragraph: “Therefore, in recent years, comprehensive school skills such as an initiative-taking approach, and self-confidence have become requirements for adaptability and success – as recognised by parents, educators, and other social factors (Liu et al.; Su & Ren, 2014)”. Ten years are recent years? And in Liu et al. the year is missing.

    • In the same paragraph: “The Ministry of Education of China has modified its educational policies to follow a more comprehensive development in academic, and various other extracurricular, activities (Chen & Chen, 2010).” No changes since 2010?

    • In the third paragraph: It has also been observed that previous studies have not placed much emphasis […] and other negative feelings, which can hinder healthy development”. Fifteen lines without any reference, you should include some in such a long explanation.

  • In the Participants and Methods section, please describe briefly the study design: you should indicate the type of study, describe better how the research respondents were recluted, include a table with the characteristics of the sample...

  • The last paragraph contains some practical implications, but it should also include the limitations of the research.

Reviewer 3 Report

Dear authors,

An interesting and well-organized article on parental rejection and adolescents' ability to learn is presented.

However, I offer some comments and suggestions for improvement.

• The summary is very complete and provides enough information about the phenomenon studied. However, I consider it important to add at least one introductory line to the topic. A summary is not usually started with the objective of the investigation, but by setting the subject according to its current situation in society.

• The keywords and the title are adjusted to the approaches and findings of the article.

• The introduction is clear and authoritative, but the references used are not current. Of approximately 29 references that make up the introduction, only 11 are current (2018-2023). I propose to improve this to make an introduction that captures the current literature and is closer to reality.

• Three hypotheses are proposed, but the objective of the investigation is not proposed. However, in the abstract if the objective is mentioned…

• The description of the sample and the procedure, although not very detailed, is sufficient. However, it is essential to have the approval of the Human Research Ethics committee. Does this study have such approval? If yes, please indicate so and include the reference number. If not, you must request it from your university.

• The instrument and its description are adequate. The data analysis process is described correctly.

• The results are described in a clear and orderly manner. However, I indicate some important issues:

- The letter N of the title of table 1 must be in italics.

- In figure 1, I think there is some data that does not look good. This figure should be reviewed.

• Discussion is not enough. Despite considering the results found, there is not enough thought about them and they are not related to previous studies.

• No prospective research is indicated.

• No limitations of the study are proposed.

• The references do not adapt to the MPDI regulations and are not current. I also consider that a low number of references are presented. Normally there are many more.

In general, the article has many shortcomings and has a lot of room for improvement. I encourage the authors to improve it according to the suggestions offered, since the article is interesting.

Round 2

Reviewer 3 Report

Dear author,

The article has improved considerably.

All suggestions made have been answered one by one.